# Atomistic Study on the Mechanical Properties of HOP–Graphene Under Variable Strain, Temperature, and Defect Conditions

**DOI:** 10.3390/nano15010031

**Published:** 2024-12-27

**Authors:** Qing Peng, Jiale Li, Xintian Cai, Gen Chen, Zeyu Huang, Lihang Zheng, Hongyang Li, Xiao-Jia Chen, Zhongwei Hu

**Affiliations:** 1School of Science, Harbin Institute of Technology, Shenzhen 518055, China; pengqing@imech.ac.cn; 2State Key Laboratory of Nonlinear Mechanics, Institute of Mechanics, Chinese Academy of Sciences, Beijing 100190, China; 23013080026@stu.hqu.edu.cn (J.L.); 22013080001@stu.hqu.edu.cn (G.C.); 23014080123@stu.hqu.edu.cn (L.Z.); lihy@stu.hqu.edu.cn (H.L.); 3Guangdong Aerospace Research Academy, Guangzhou 511458, China; 4Institute of Manufacturing Engineering, Huaqiao University, Xiamen 361021, China; 5Institute of Mechanical Engineering and Automation, Huaqiao University, Xiamen 361021, China; 6School of Mechanical Engineering, Hubei University of Technology, Wuhan 430068, China; 7Hubei Key Laboratory of Electronic Manufacturing and Packaging Integration, Wuhan University, Wuhan 430072, China

**Keywords:** HOP–graphene, molecular dynamics, mechanical properties, defects

## Abstract

HOP–graphene is a graphene structural derivative consisting of 5-, 6-, and 8-membered carbon rings with distinctive electrical properties. This paper presents a systematic investigation of the effects of varying sizes, strain rates, temperatures, and defects on the mechanical properties of HOP–graphene, utilizing molecular dynamics simulations. The results revealed that Young’s modulus of HOP–graphene in the armchair direction is 21.5% higher than that in the zigzag direction, indicating that it exhibits greater rigidity in the former direction. The reliability of the tensile simulations was contingent upon the size and strain rate. An increase in temperature from 100 K to 900 K resulted in a decrease in Young’s modulus by 7.8% and 2.9% for stretching along the armchair and zigzag directions, respectively. An increase in the concentration of introduced void defects from 0% to 3% resulted in a decrease in Young’s modulus by 24.7% and 23.1% for stretching along the armchair and zigzag directions, respectively. An increase in the length of rectangular crack defects from 0 nm to 4 nm resulted in a decrease in Young’s modulus for stretching along the armchair and zigzag directions by 6.7% and 5.7%, respectively. Similarly, an increase in the diameter of the circular hole defect from 0 nm to 4 nm resulted in a decrease in Young’s modulus along both the armchair and zigzag directions, with a corresponding reduction of 11.0% and 10.4%, respectively. At the late stage of tensile fracture along the zigzag direction, HOP–graphene undergoes a transformation to an amorphous state under tensile stress. Our results might contribute to a more comprehensive understanding of the mechanical properties of HOP–graphene under different test conditions, helping to land it in potential practical applications.

## 1. Introduction

Graphene, a material that people first synthesised in 2004, has properties such as very high electron mobility, excellent thermal conductivity, superior mechanical strength and flexibility, and large specific surface area, which have led to a wide range of applications in fields such as electric equipment, biomedicine, optoelectronic devices, and composite materials [1,2,3,4]. Given the favourable characteristics of graphene and its potential applications, there is a growing interest among scientists in graphene derivatives [5,6,7,8,9,10,11,12,13,14,15,16,17,18,19,20,21,22,23,24,25,26]. However, challenges remain in the areas of material preparation and experimental techniques. Consequently, many researchers have turned to numerical simulation methods to investigate novel two-dimensional carbon materials [27,28,29,30,31,32,33,34,35,36,37,38]. Sun et al. [39] used first-principles DFT calculations to simulate the stress–strain response of 11 different graphene isomers under uniaxial and biaxial stretching and found that the graphene isomers possessed a high Young’s modulus and a high ultimate tensile strength along with low density. Xie et al. [40] investigated the effect of different stretching angles on the mechanical properties of PSI-graphene by molecular dynamics. Xiao et al. [41] investigated the mechanical properties of biphenylene without defects and with various shapes and sizes of hollow defects under uniaxial stretching by molecular dynamics simulation, revealing that Young’s modulus and strength depend on the area of the defects, whereas the breaking strain and toughness depend on the shapes of the holes.

HOP–graphene is a recently discovered two-dimensional carbon allotrope derived from the combination of hexagons, octagons, and pentagons, which gives it its distinctive name. Mandal et al. [42] predicted it by first-principles calculations and investigated its electronic structure and transport properties. Their results show that HOP–graphene is an energetically stable 2D carbon isomer with metallic behaviors, and its 1D derivatives have properties such as NDR phenomena in electron transport, current rectification properties, etc. Sui et al. [43] employed molecular dynamics to examine nine graphene isomers, including HOP–graphene, and discovered a pronounced correlation between the tensile mechanical properties of graphene isomers and their structural morphology and tensile orientation.

However, a systematic exploration of the mechanical aspects of this material is still lacking, despite its importance in applications where complex mechanical loadings are unavoidable. Therefore, the objective of this study is to examine the mechanical characteristics of HOP–graphene through molecular dynamics simulations. The investigation will encompass a comprehensive analysis of the influences of anisotropy, size, strain rate, temperature, vacancy defects, rectangular defects, and circular defects. Additionally, the mechanical response of HOP–graphene under diverse conditions will be elucidated. This study enhances our comprehension of the characteristics of HOP–graphene, which may inform future applications in mechanical properties and defect failure.

## 2. Materials and Methods

Molecular dynamics simulations were conducted using LAMMPS (lammps-23Jun2022) to investigate the mechanical properties of HOP–graphene structures. LAMMPS is a software dedicated to the simulation of large-scale atomic systems and is widely used for its efficient kinetic simulation capabilities [44]. In addition, the original structure of HOP graphene was constructed using OVITO software (version 3.7.10), and the structural changes under stress were visualised and studied using OVITO [45].

Figure 1a shows the structural features of HOP–graphene, which contains 5-, 6-, and 8-membered carbon rings with 10 carbon atoms in each protocell. Similar to graphene, HOP–graphene is a monolayer nanomaterial with a thickness of 0.334 nm. The lattice constants along the armchair and zigzag directions are a = 0.5647 nm and b = 0.4908 nm, respectively, with γ = 89.5519°. Additionally, the length of the C-C bond is observed to fall within the range of 0.139 nm to 0.156 nm. As shown in Figure 1c, we constructed a model consisting of 20,160 carbon atoms with a length of 23.7 nm, a width of 23.6 nm, and a vacuum layer of 10 nm along the z-axis, thus ensuring the reliability of the simulation results.

To mimic the behavior of the monolayer material in a realistic setting, we constructed 3D simulation dimensions with periodic boundary conditions in the x- and y-axes and free boundary conditions applied in the z-axis. The system temperature was set to 300 K, and the pressure was set to zero. To capture the atomic interactions occurring during the course of the simulation, a timestep of 0.001 ps was employed. The AIREBO-M potential function was selected for use with HOP–graphene, as it accounts for interactions between multiple atoms and is particularly well-suited to the study of carbon-based materials [46,47,48], especially the mechanical properties, including fracture [49].

We performed energy minimisation using the conjugate gradient (cg) technique to stabilise the structure. We then relaxed the system for 40 ps to ensure the system becomes thermodynamic equilibrium with an isothermal–isobaric (NPT) ensemble. Throughout the stretching process, the pressure perpendicular to the stretching direction was kept at zero and the system temperature was stabilised at the target value. We calculated and recorded the stress and strain data using the built-in calculation commands of LAMMPS, including the stress component for each atom, and the average stress and von Mises stress. Von Mises stress is a key indicator for assessing the yield strength of materials and is widely used in engineering design and evaluation, which has a significant impact on ensuring the safety and reliability of materials and structures.

Ultimately, the stress–strain curves were generated by fitting the stress–strain data, and the values of the fundamental mechanical properties, including fracture strain (ε), failure strength (σ), Young’s modulus (E), and toughness, were extracted from the curves to facilitate a more comprehensive investigation of the mechanical behaviors of HOP–graphene in uniaxial stretching under diverse conditions. Specifically, the fracture strain indicates the strain value at the breaking point of the material in the tensile test, which is equivalent to the strain value at the maximum stress. The failure strength refers to the maximum stress value when the material reaches the state of destruction or failure in the process of force. Young’s modulus is a physical quantity that describes the stiffness of the material within the elasticity range, and it is defined as the ratio of the material in the ratio between the stress and the corresponding strain. The toughness reflects the ability to absorb the energy of the material under the action of tensile force, which is expressed by the area under the stress–strain curve.

By using MD modelling, we were able to efficiently simulate the uniaxial stretching of HOP–graphene and identify important mechanical property factors from the stress–strain curves. These findings are important for improving our understanding and application of graphene materials.

## 3. Results

### 3.1. Size Effect

Molecular dynamics simulations are based on statistical mechanics, but the limited number of atoms in small systems may lead to less reliable statistical results for accurate prediction of mechanical properties, while large systems increase computational costs. Therefore, considering size effects is important in MD modelling and helps to use computational resources efficiently. Appropriate choice of model dimensions can ensure that the simulation results accurately reflect the behaviors of the material at a particular scale, thus improving its realism and reliability.

We investigated four defect-free HOP–graphene models ranging in size from 11.9 nm × 11.7 nm to 33.9 nm × 33.3 nm, with model sizes increasing in double and the number of atoms in the system ranging from 2550 to 40,800. To investigate the influence of size on mechanical properties, uniaxial tensile simulations were conducted on these models along the armchair and zigzag directions. Figure 2 and Figure 3 show the stress–strain curves of HOP–graphene in the armchair and zigzag directions, respectively, as well as the results of calculations of fracture strain, failure strength, Young’s modulus, and toughness as a function of temperature. The curves of HOP–graphene of different sizes stretched along the armchair and zigzag directions before fracture are basically the same, indicating that the effect of size on Young’s modulus is small. However, there is a significant difference between the fracture strain and the failure strength. As illustrated in Figure 2, when the atomic number is increased from 2550 to 40,800, the fracture strain along the armchair direction decreases by 7.07% (from 0.009 to 0.092), while the failure strength decreases by 4.24% (from 75.17 GPa to 71.98 GPa). As illustrated in Figure 3, the fracture strain along the zigzag direction decreases by 7.79% (from 0.077 to 0.071), while the failure strength decreases by 5.38% (from 49.93 GPa to 47.25 GPa).

The HOP–graphene model with 2550 atoms shows the largest difference in fracture stress–strain behavior under uniaxial force, and the stress–strain curves gradually converge as the number of atoms in the system increases. The curves almost intersect at atom numbers of 10,200 and above, a trend that suggests that the simulation results should tend to stabilize and converge when the model size rises to a reasonable level. As a compromise of accuracy and computational resources, we choose a model with 20,160 atoms in the subsequent study.

### 3.2. Strain Rate Effect

The strain rate is the rate at which a material changes strain per unit of time. Some materials’ properties, including mechanical properties, are sensitive to the strain rate. To check the strain rate effect on the mechanical properties of HOP–graphene, a total of four distinct strain rates were investigated, which represents a typical range of strain rates employed in two-dimensional material tensile simulations [50].

Figure 4 illustrates the stress–strain curves of HOP–graphene under disparate strain rate conditions for uniaxial stretching along the armchair and zigzag directions, respectively. The figure demonstrates that the mechanical response of HOP–graphene in uniaxial stretching simulations is influenced by the strain rate. As the strain rate increases from 1 × 10^8^ s^−1^ to 5 × 10^9^ s^−1^ along the armchair direction, the fracture strain rises from 0.083 to 0.092, representing a 10.84% increase, while the failure strength increases from 68.03 GPa to 71.83 GPa or 5.58%. In the zigzag direction, the fracture strain increased from 0.067 to 0.083, representing a 23.88% increase, while the failure strength increased from 45.54 GPa to 51.21 GPa or 12.45%. The stress–strain curves at varying strain rates exhibit minimal divergence prior to the onset of fracture, suggesting that the strain rate has a negligible impact on Young’s modulus.

As the strain rate increases, the strain at fracture and the breaking strength also increases, indicating enhanced ductility. Conversely, a reduction in the strain rate results in a decrease in both the strain at fracture and the breaking strength, indicating an increase in brittleness. This is due to the fact that at low strain rates, the material has more time for energy balance and dislocation motion. However, these mechanisms may not be apparent for highly brittle or defect-sensitive materials. Conversely, low strain rates may result in the formation of defects and cracks that are more likely to expand, thereby reducing strength and fracture strain and exhibiting brittle characteristics.

It is worth noting that the difference between the results at strain rates of 1 × 10^9^ s^−1^ and 5 × 10^8^ s^−1^ is very small: the difference in fracture strain in the armchair direction is 4.26%, and the difference in failure strength is 2.70%; the difference in fracture strain in the zigzag direction is 0.16%, and the difference in failure strength is 0.21%. The near-unanimous results indicate that HOP–graphene possesses stable plastic deformation characteristics and reasonable yield strength at these strain rates, providing reliable data for the mechanical properties of the material. We chose a strain rate of 1 × 10^9^ s^−1^ for the subsequent simulations.

### 3.3. Orientation Dependent Mechanical Properties

In practical production applications, HOP–graphene may be subjected to stresses in different directions, and the anisotropy might cause HOP–graphene to exhibit different mechanical properties when stretched along different directions. The mechanical properties of HOP–graphene along the armchair and zigzag are shown in Table 1. In addition, we performed uniaxial tensile simulation tests on graphene using the same parameter settings to compare and analyse the mechanical properties of HOP–graphene and graphene.

The stress–strain curves of uniaxial stretching of HOP–graphene and graphene along the armchair and zigzag directions, respectively, are shown in Figure 5. The fracture strain of HOP–graphene along the armchair and zigzag directions are 0.095 and 0.071, respectively, which is 33.8% higher in the armchair direction than in the zigzag direction, and that of graphene is 29.1% lower in the armchair direction than in the zigzag direction. The failure strength of HOP–graphene along the armchair and zigzag directions are 73.1 GPa and 46.9 GPa, respectively, which were 55.9% higher in the armchair direction than in the zigzag direction, and that of graphene is 16.7% lower in the armchair direction than in the zigzag direction. The Young’s modulus of HOP–graphene along the armchair and zigzag directions are 1132 GPa and 932 GPa, respectively, which is 21.5% higher in the chair direction than in the zigzag direction, and that of graphene is 13.2% lower in the armchair direction than in the zigzag direction. The toughness of HOP–graphene along the armchair and zigzag directions are 4.11 J/m^3^ and 1.91 J/m^3^, respectively, which is 115.2% higher in the armchair direction than in the zigzag direction, and that of graphene is 40.3% lower in the armchair direction than in the zigzag direction.

HOP–graphene has better mechanical properties in the chair orientation than in the zigzag orientation. Overall, it appears that graphene has better mechanical properties, probably due to the unique arrangement of carbon atoms in HOP–graphene, which differs from the regular hexagonal structure of graphene. We introduce the concept of surface area density to further explain the reason for the difference in their mechanical properties. The mass surface density directly affects the mechanical properties of materials and can, to a certain extent, reflect the structural properties of 2D materials [50].

The mass surface density of the two-dimensional carbon material can be calculated using the following Equation (1):(1)$$\rho =\frac{m}{s},$$
where m—the total mass of carbon atoms in the region and s—the area of the region in which a certain number of carbon atoms are located.

The mass surface density of HOP–graphene is calculated by Equation (1) to be 7.19 × 10^−7^ kg/m^2^, while that of graphene is calculated to be 7.60 × 10^−7^ kg/m^2^. It can be observed that the mass surface density of graphene is 5.7% higher than that of HOP–graphene, which indicates that graphene exhibits superior mechanical properties.

On this basis, the projected line density can be calculated using the following Equation (2):(2)$$\left\{\begin{array}{c} \rho_{i}=\rho_{ij}\times l_{j}\\ \rho_{j}=\rho_{ij}\times l_{i} \end{array}\right.,$$
where $\rho_{ij}$—mass surface density, $\rho_{i}$—projected line density of the material along the i-direction, $\rho_{j}$—projected line density of the material along the j-direction, $l_{i}$—the length of the projection of the region along the i-direction, and $l_{j}$—the length of the projection of the region along the j-direction.

The projected line densities of HOP–graphene along the armchair and zigzag directions can be calculated from Equation (2) as 4.06 × 10^−16^ kg/m and 3.53 × 10^−16^ kg/m, respectively. This reflects the fact that the armchair direction has a greater fracture strain, failure strength, Young’s modulus, and tensile toughness. These findings are consistent with those obtained from the simulations. The concept of projected line density provides a partial explanation for the differences in the mechanical properties of HOP–graphene in the two directions.

### 3.4. Temperature Effect

Materials research and engineering rely on understanding the mechanical properties of materials at high temperatures. It is essential to conduct a comprehensive investigation into the relationship between temperature and mechanical properties to guarantee the effective utilisation of HOP–graphene in extreme temperature conditions.

The fracture stress of two-dimensional materials varies with temperature and is usually described by the Arrhenius equation or the thermal activation model (3):(3)$$\sigma_{c}(T)=\sigma_{0}exp\left(-\frac{E_{a}}{k_{B}T}\right),$$
where $\sigma_{c}(T)$—the temperature fracture strength at T, $\sigma_{0}$—the fracture strength at zero temperature, $E_{a}$—the activation energy of interatomic bonds at the crack tip, $k_{B}$—Boltzmann constant, T—the absolute temperature. At high temperatures, thermal vibrations weaken the interatomic bond strength, reducing fracture strength.

To elucidate the precise impact of temperature on the simulation outcomes, we conducted uniaxial tensile simulations on defect-free HOP–graphene at 200 K intervals within the temperature range of 100 K to 900 K. Figure 6 and Figure 7 show the stress–strain curves of HOP–graphene under stretching along the armchair and zigzag directions, respectively, as well as the calculated results of fracture strain, failure strength, Young’s modulus, and toughness at different temperatures. The mechanical properties of HOP–graphene exhibit a decrease with increasing temperature.

As shown in Figure 6, during the stretching process in the armchair direction, the temperature increased from 100 K to 900 K. The fracture strain of HOP–graphene decreased from 0.120 to 0.054, with a decrease of 55.00%; the failure strength decreased from 84.0 GPa to 44.0 GPa, with a decrease of 47.62%; the Young’s modulus decreased from 1142 GPa to 1053 GPa, with a decrease of 7.79%; and toughness decreased from 6.35 J/m^3^ to 1.40 J/m^3^, with a decrease of 77.95%. As shown in Figure 7, during zigzag direction stretching, the fracture strain of HOP–graphene decreases from 0.102 to 0.017, with a decrease of 83.33%; the failure strength decreases from 59.5 GPa to 13.4 GPa, with a decrease of 77.50%; the Young’s modulus decreases from 939 GPa to 912 GPa, with a decrease of 2.88%; the toughness decreases from 3.65 J/m^3^ to 0.12 J/m^3^, with a decrease of 96.71%. These changes are attributed to the enhanced thermal vibration of atoms in graphene at high temperatures and the intensified positional fluctuations of the atoms, which affect the overall strength and toughness of the material, thus making the mechanical properties of graphene decrease under high temperature conditions [51,52,53].

### 3.5. The Vacancy Defects Effect

Normally, atoms in a crystal are arranged in a certain periodic structure. However, in actual materials, various defects that deviate from the ideal state often appear in the crystal lattice. Defects are an inherent consequence of the synthesis and utilization of HOP–graphene. These defects have the potential to exert a considerable influence on the material’s electrical, thermal, and mechanical properties [54,55,56,57,58,59,60,61,62]. Vacancy defects are a common type of defect in the lattice structure of solid materials, usually caused by missing atoms or ions at lattice positions. During the processing and application of materials, vacancy defects may lead to increased brittleness and decreased fatigue properties. In engineering, reasonable control of the formation and distribution of vacancy defects can improve material properties. Especially in semiconductor materials, the modulation of vacancy defects can help to adjust the electronic properties of the materials to fabricate more efficient electronic devices. Therefore, the study of vacancy defects is not only crucial to materials science and engineering but also contributes to a deeper understanding of the behaviors and properties of materials in practical applications [63,64].

A series of simulations were conducted in which 0.1%, 0.5%, 1%, 1.5%, 2%, 2.5%, and 3% of atoms were randomly removed from a system containing 20,160 carbon atoms. The objective was to generate vacancy defects and compare them with the defect-free HOP–graphene. This was conducted to investigate the effect of defects on the mechanical properties of HOP–graphene. Figure 8 and Figure 9 show the stress–strain curves, the four mechanical properties, and the coefficient of variation of the mechanical properties with void concentration for HOP–graphene during uniaxial stretching in the armchair and zigzag directions, respectively, at different concentrations of void defects. The coefficient of variation is the ratio of the standard deviation to the mean and indicates the relative degree of volatility of the data, with larger coefficients of variation indicating greater volatility of the data.

With the increase in vacancy defect concentration, the above four mechanical properties showed a decreasing trend. As shown in Figure 8, when the concentration of vacancy defects increased from 0% to 3% along the direction of the armchair, the fracture strain of HOP–graphene decreased from 0.095 to 0.048, with a decrease of 49.47%; the failure strength decreased from 73.1 GPa to 33.9 GPa, with a decrease of 53.63%; Young’s modulus decreased from 1132 GPa to 852 GPa, with a decrease of 24.73%; and the toughness decreased from 4.1 J/m^3^ to 0.96 J/m^3^, with a decrease of 76.59%. As shown in Figure 9, along the zigzag direction, the fracture strain of HOP–graphene decreases from 0.07 to 0.019, with a decrease of 72.86%; the failure strength decreases from 46.9 GPa to 12.5 GPa, with a decrease of 73.35%; Young’s modulus decreases from 932 GPa to 717 GPa, with a decrease of 23.07%; the toughness decreases from 1.91 J/m^3^ to 0.13 J/m^3^, a decrease of 93.19%. The coefficients of variation of Young’s modulus were 9.6% and 9.7% for stretching along the armchair and zigzag directions, respectively, which were much lower than the other three mechanical properties, suggesting that the effect on the stiffness of the HOP–graphene was smaller at lower vacancy concentrations.

### 3.6. Defects Effect

In practice, HOP–graphene may be affected by external factors such as mechanical stress and chemical corrosion, resulting in the formation of different types of defects such as cracks and holes. These defects are the main source of stress concentration, which causes stress concentration locally in the material and accelerates the occurrence of fracture. The size and shape of cracks and holes directly affect their propagation direction and velocity in the material, which in turn affects the overall mechanical properties. Therefore, studying the shape and size of defects is crucial to the mechanical properties of HOP–graphene. Such studies not only help to gain a deeper understanding of how defects affect the behaviors of materials but also provide important guidance and optimization ideas for material design and engineering applications.

In the models in Section 3.6.1 and Section 3.6.2 below, we set up rectangular and circular defects of different sizes to investigate their effect on mechanical properties. Figure 10 shows the von Mises stress distribution, which visualises the distribution of stress and the development of fracture during stretching. The color of the atoms corresponds to the von Mises stress values, reflecting the stress distribution in the material during tensile deformation. The first figure of each data set shows the unstretched model after relaxation is completed; the second figure presents the morphology of the model about to fracture; the third figure shows the crack extension process; and the fourth figure shows the region of transition to amorphous state. In this case, the large red area indicates the presence of an amorphous structure, at which point the von Mises stress has reached its limit.

#### 3.6.1. Rectangular Crack Defects

A rectangular crack defect perpendicular to the stretching direction was set up in the center of the model, and the effect of four rectangular cracks of width 1 nm and lengths 1, 2, 3, and 4 nm on the HOP–graphene was investigated. Uniaxial tensile simulation experiments were carried out in the armchair and zigzag directions, and the results were compared to the HOP–graphene without defects. Figure 11a,b shows the stress–strain curves in the presence of rectangular cracks of different lengths. Figure 12a,b shows the four mechanical properties corresponding to Figure 11a,b. As the crack length increases, the fracture strain and failure strength decrease gradually. When the crack length was increased from 0 nm to 4 nm and stretched along the armchair direction, the fracture strain of HOP–graphene decreased from 0.095 to 0.033, with a decrease of 65.26%; the failure strength decreased from 73.1 GPa to 32.0 GPa, with a decrease of 56.22%; Young’s modulus decreased from 1132 GPa to 1056 GPa, with a decrease of 6.71%; the toughness decreased from 4.11 J/m^3^ to 0.62 J/m^3^, with a decrease of 84.91%. Along the zigzag direction, the fracture strain of HOP–graphene decreases from 0.071 to 0.358, with a decrease of 49.30%; the failure strength decreases from 46.9 GPa to 25.7 GPa, with a decrease of 45.20%; Young’s modulus decreases from 932 GPa to 879 GPa, with a decrease of 5.69%; the toughness decreases from 1.91 J/m^3^ to 0.52 J/m^3^, with a decrease of 72.77%. Figure 10a,b shows the von Mises stress distribution during stretching of HOP–graphene at a crack length of 3 nm. The presence of cracks leads to higher stress concentration and faster crack extension, especially at longer cracks. Amorphous phases appear during stretching along the zigzag direction, which does not occur during stretching along the armchair direction.

#### 3.6.2. Circular Void Defects

Circular hollow defects with diameters of 1, 2, 3, and 4 nm were placed in the center of the model, respectively, to investigate their effects on the HOP–graphene. Uniaxial tensile simulation experiments were carried out in the armchair and zigzag directions, and the results were compared with the HOP–graphene without defects. Figure 11c,d shows the tensile stress–strain curves of HOP–graphene in the presence of circular defects with different diameters. Figure 12c,d shows the four mechanical properties corresponding to Figure 11c,d. Circular hollow defects with larger diameters exhibit earlier strength loss and lower fracture strain points. When the diameter of circular hollow defects increases from 0 nm to 4 nm, the fracture strain of HOP–graphene decreases from 0.095 to 0.037 in the direction of armchair, a decrease of 61.05%; the failure strength decreases from 73.1 GPa to 34.4 GPa, a decrease of 52.94%; Young’s modulus decreases from 1132 GPa to 1008 GPa, a decrease of 10.95%; the toughness decreased from 4.11 J/m^3^ to 0.74 J/m^3^, a decrease of 82.00%. Stretching along the zigzag direction, the strain at break of HOP–graphene decreased from 0.071 to 0.042, with a decrease of 40.84%; the failure strength decreased from 46.9 GPa to 28.7 GPa, with a decrease of 38.81%; Young’s modulus decreased from 932 GPa to 835 GPa, with a decrease of 10.41%; the toughness decreased from 1.91 J/m^3^ to 0.79 J/m^3^, a decrease of 58.64%. Figure 10c,d shows the von Mises stress distribution of HOP–graphene with a hole diameter of 3 nm during stretching. Similar to rectangular cracks, circular defects also lead to stress concentration at their edges. The fracture starts from the edge of the defect and extends along the stress concentration region. Larger diameter circular defects induce more significant stress concentrations, which have a significant negative impact on the strength and toughness of the material. Overall, an increase in the diameter of circular defects significantly affects the stress–strain behaviour and fracture mode of HOP–graphene, with amorphous phases appearing during stretching in the zigzag direction, as in the case of rectangular cracks.

#### 3.6.3. Radial Distribution Function

The amorphous phase is a state of solid matter in which the atoms or molecules within the substance lack a long-range ordered arrangement, existing instead in a disordered, random distribution. Given that the amorphous phase was observed during the tensile fracture of HOP–graphene along the zigzag direction, a further analysis was conducted using the radial distribution function (RDF). The RDF describes the distribution of particles around a reference particle, reflecting the spatial relationship between the particles. It was found that the amorphous phases lack an obvious periodic structure. The RDF thus provides an explanation of the structural changes that occur when HOP–graphene undergoes tensile fracture, resulting in the formation of the amorphous phases.

As shown in Figure 13, the RDF data was subjected to analysis for samples exhibiting rectangular defects with a length of 3 nm and circular defects with a diameter of 3 nm, which were stretched along the zigzag direction. The initial data set reflects the structure at the outset of the experiment; the second data set reflects the structure prior to the occurrence of fracture; the third data set reflects the structure at the onset of crack development; and the subsequent three data sets illustrate the evolution of the amorphous phase. It was observed that as the strain increased, the peak position on the RDF curve shifted to the right and the peak height decreased. Additionally, the valley between the peaks gradually filled up. Prior to rupture, the atomic bond length is elongated under stretching, which causes the peak position to shift to the right. Following rupture but prior to the amorphous phase, the peak position remains relatively unchanged. After the amorphous phase has appeared, the atomic bonds begin to break, which results in a notable shift of the peak position to the right and a reduction in the peak height. This suggests that the amorphous phase emerges during the late stages of stretching HOP–graphene along the zigzag direction.

## 4. Conclusions

In this study, molecular dynamics simulations were employed to examine the influence of size, strain rate, temperature, vacancy defects, and rectangular and circular defects on the mechanical properties of HOP–graphene. The findings indicate that the mechanical properties exhibit a tendency towards stabilisation when the strain rate is below 1 × 10⁹ s^−1^. The mechanical properties of HOP–graphene were compared with those of graphene, and it was found that the mechanical properties of HOP–graphene were higher than those of zigzag along the armchair direction, while the opposite was true for graphene. Defect-free HOP–graphene was subjected to tensile testing along the armchair and zigzag directions at a temperature of 300 K and a strain rate of 1 × 10^9^ s^−1^. The Young’s modulus was found to be 1132 GPa and 932 GPa, respectively, and a significant difference in the mechanical properties was observed between the two directions, with the former exhibiting higher stiffness.

It has been demonstrated that alterations in temperature during the stretching process exert a considerable influence on the mechanical properties of HOP–graphene. A reduction in temperature from 100 K to 900 K has been observed to result in a decrease in Young’s modulus, from 1142 GPa to 1053 GPa (7.79%) for stretching along the armchair direction, and from 939 GPa to 912 GPa (2.88%) for stretching along the zigzag direction. The introduction of vacancy defects results in a deterioration of the mechanical properties of HOP–graphene, with a concentration of local strain occurring in the vicinity of the defects. An increase in vacancy defects from 0% to 3% resulted in a reduction in Young’s modulus from 1132 to 852 GPa (24.73%) when the material was stretched along the armchair direction, and from 932 to 717 GPa (23.07%) when it was stretched along the zigzag direction.

The mechanical properties of HOP–graphene are also significantly influenced by defects of varying sizes and shapes. As the length of the rectangular crack increased to 4 nm, a decrease in Young’s modulus was observed, from 1132 GPa to 1056 GPa (6.71%) when stretched along the armchair direction, and from 932 GPa to 879 GPa (5.69%) when stretched along the zigzag direction. As the diameter of the circular hole defects increased to 4 nm, a decrease in Young’s modulus was observed, from 1132 GPa to 1008 GPa (11.0%) when the material was stretched along the armchair direction, and from 932 GPa to 835 GPa (10.4%) when the material was stretched along the zigzag direction. The amorphous phase observed in both rectangular and circular defects when stretched along the zigzag direction is not present when the same process is carried out along the armchair direction.

The findings of our molecular dynamics study offer a detailed, atomic-scale understanding of the mechanical behaviors of HOP–graphene across a range of parameters, providing valuable insights that will inform its diverse applications in nanotechnology and materials engineering. The thermal conductivities, mass-transport properties, and applications in batteries and energy fields are among the interesting research topics.

## Figures and Tables

**Figure 1 nanomaterials-15-00031-f001:**
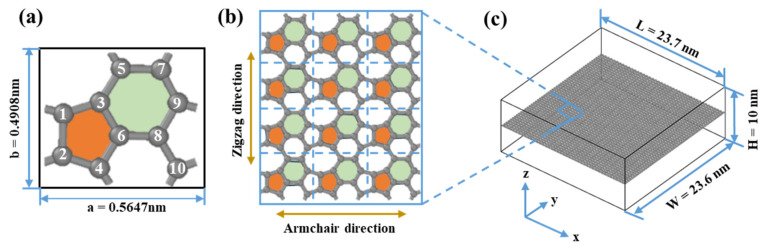
Structural schematic of HOP–graphene. (**a**) Primitive unit cell of HOP–graphene. (**b**) Monolayer HOP–graphene. (**c**) 3D view of HOP–graphene with 20,160 atoms. The numbers 1–10 indicate the 10 atoms in the primitive unit cell of HOP-graphene, and the orange and green colours denote the five- and six-membered rings, respectively.

**Figure 2 nanomaterials-15-00031-f002:**
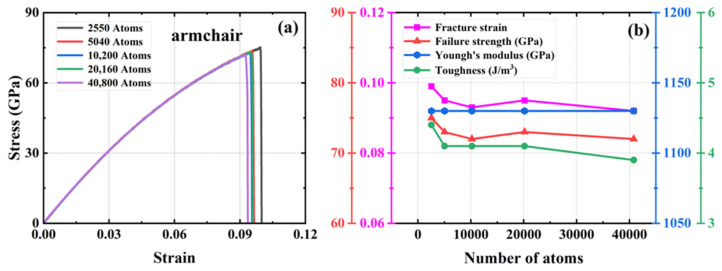
Effect of different sizes on the mechanical properties of HOP–graphene by stretching along the armchair direction. (**a**) Stress–strain curves; (**b**) Variation of four mechanical properties with different sizes.

**Figure 3 nanomaterials-15-00031-f003:**
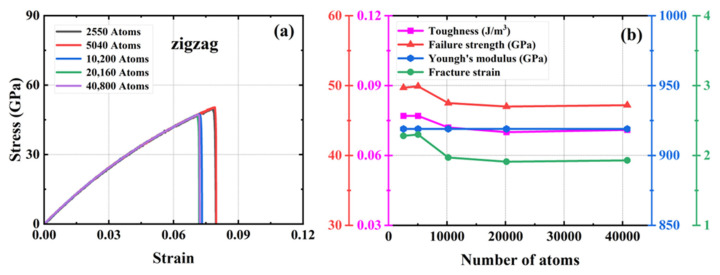
Effect of different sizes on the mechanical properties of HOP–graphene by stretching along the zigzag direction. (**a**) Stress–strain curves; (**b**) Variation of four mechanical properties with different sizes.

**Figure 4 nanomaterials-15-00031-f004:**
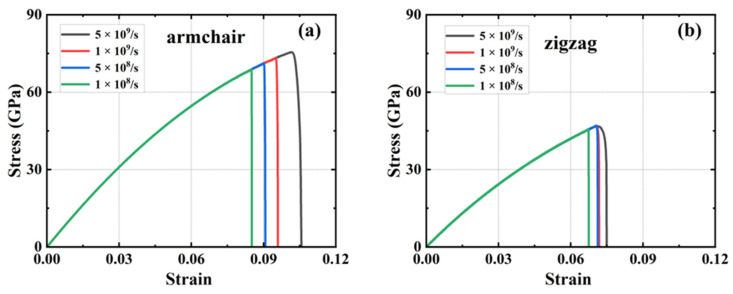
Stress–strain curves at different strain rates. HOP–graphene stretching (**a**) along the armchair direction; (**b**) along the zigzag direction.

**Figure 5 nanomaterials-15-00031-f005:**
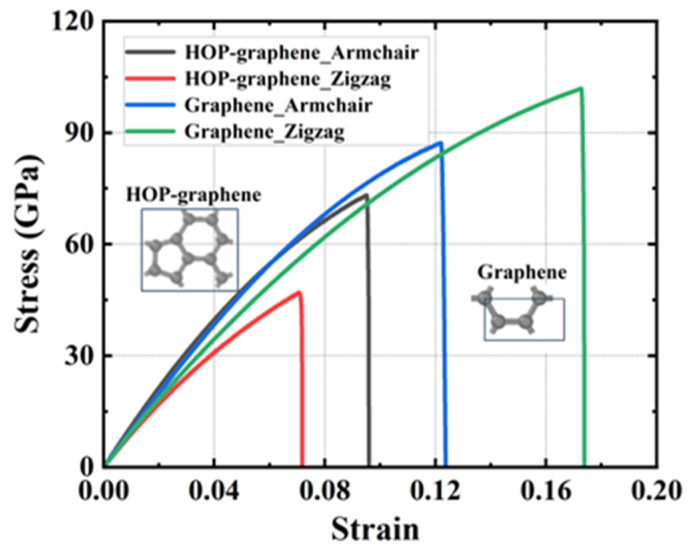
Stress–strain curves of HOP–graphene and graphene stretched in different directions.

**Figure 6 nanomaterials-15-00031-f006:**
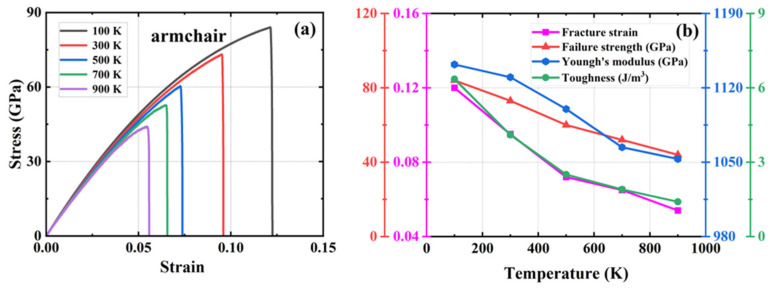
Effect of different temperatures on the mechanical properties of HOP–graphene by stretching along the armchair direction. (**a**) Stress–strain curves; (**b**) Variation of four mechanical properties with different temperatures.

**Figure 7 nanomaterials-15-00031-f007:**
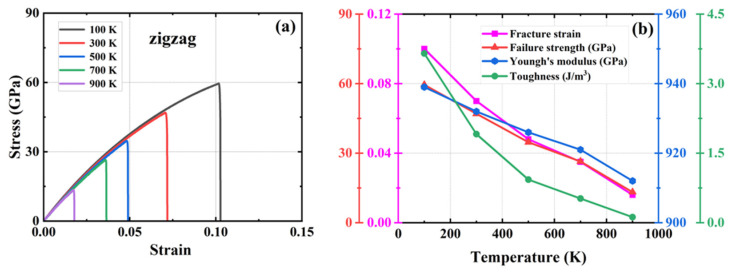
Effect of different temperatures on the mechanical properties of HOP–graphene by stretching along the zigzag direction. (**a**) Stress–strain curves; (**b**) Variation of four mechanical properties with different temperatures.

**Figure 8 nanomaterials-15-00031-f008:**
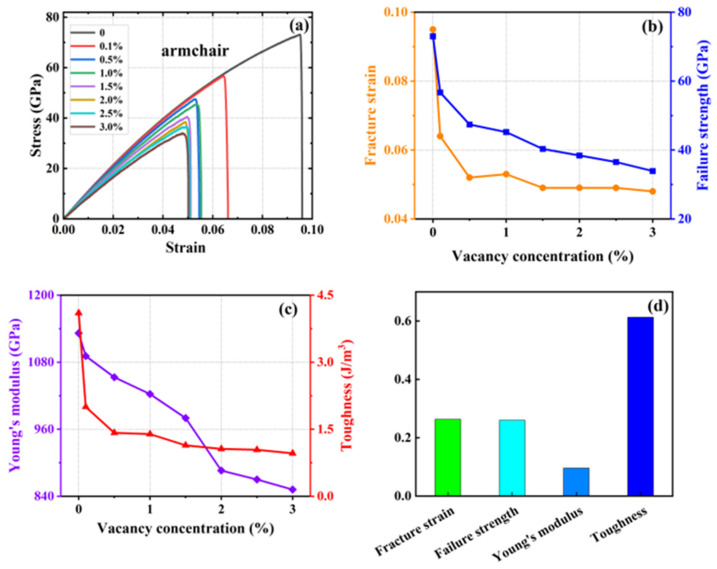
Effect of different vacancies on the mechanical properties of HOP–graphene by stretching along the armchair direction. (**a**) Stress–strain curves; (**b**,**c**) Variation of four mechanical properties with different vacancy concentrations; (**d**) Coefficient of variation of mechanical properties with vacancy concentration.

**Figure 9 nanomaterials-15-00031-f009:**
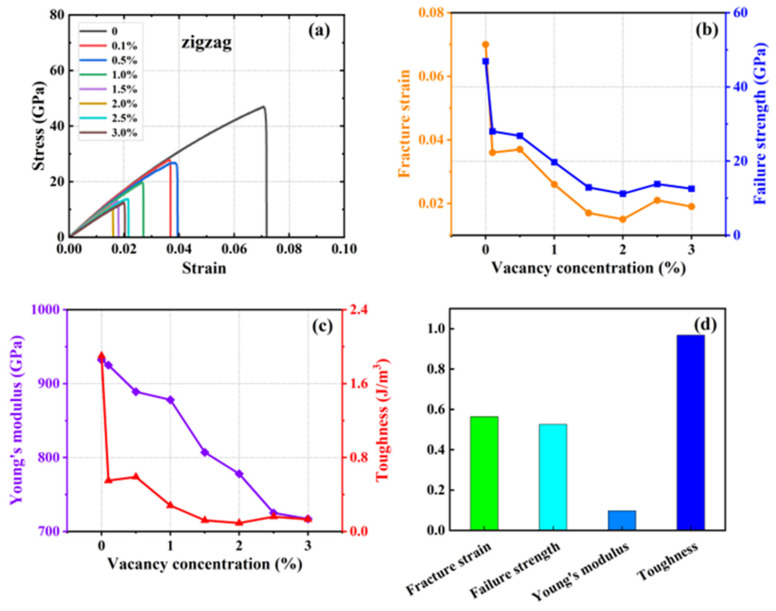
Effect of different vacancy on the mechanical properties of HOP–graphene by stretching along the zigzag direction. (**a**) Stress–strain curves; (**b**,**c**) Variation of four mechanical properties with different vacancy concentrations; (**d**) Coefficient of variation of mechanical properties with vacancy concentration.

**Figure 10 nanomaterials-15-00031-f010:**
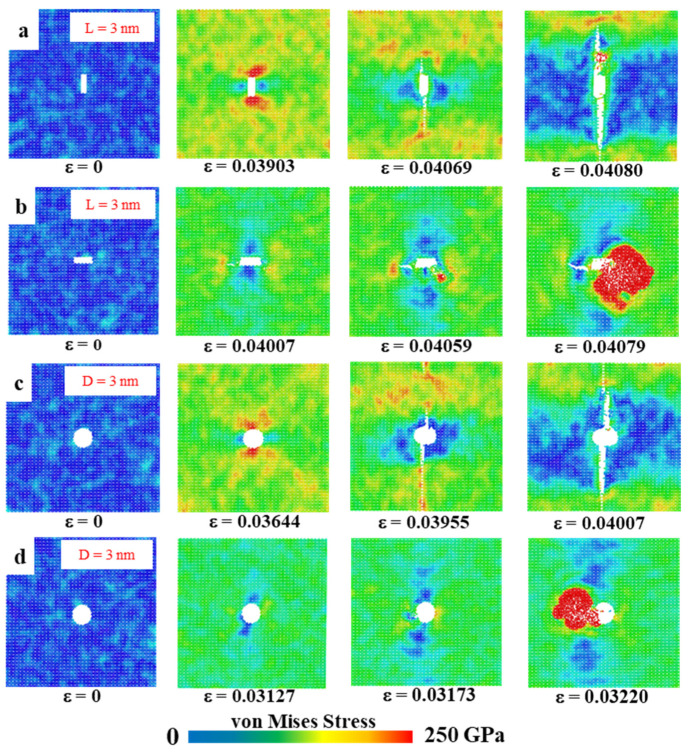
Von Mises stress distribution in HOP–graphene with various defects. (**a**,**b**) Von Mises stresses for rectangular crack defects of 3 nm length, stretched along the armchair and zigzag directions, respectively; (**c**,**d**) Von Mises stresses for 3 nm circular hole defects by stretching along the armchair and zigzag directions, respectively.

**Figure 11 nanomaterials-15-00031-f011:**
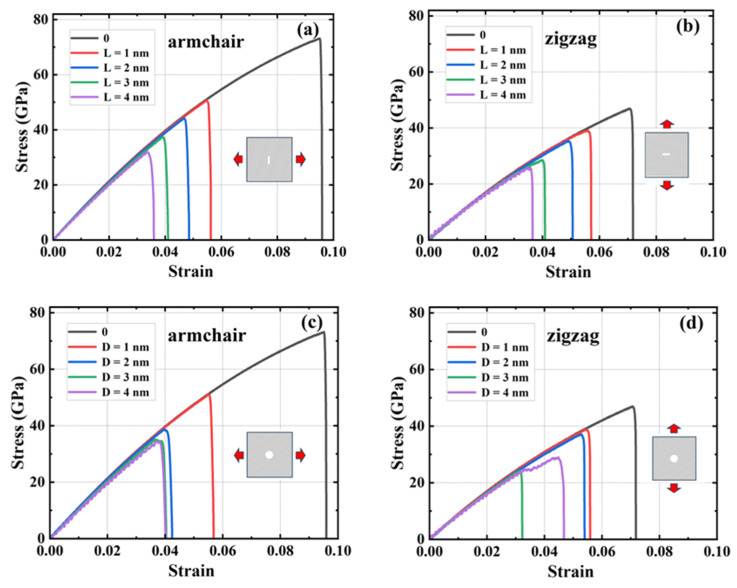
Stress–strain curves of HOP–graphene with rectangular cracks of different lengths under uniaxial tension (**a**) in the armchair direction and (**b**) in the zigzag direction. Stress–strain curves of HOP–graphene with circular cracks of different diameters under uniaxial tension (**c**) in the armchair direction and (**d**) in the zigzag direction.

**Figure 12 nanomaterials-15-00031-f012:**
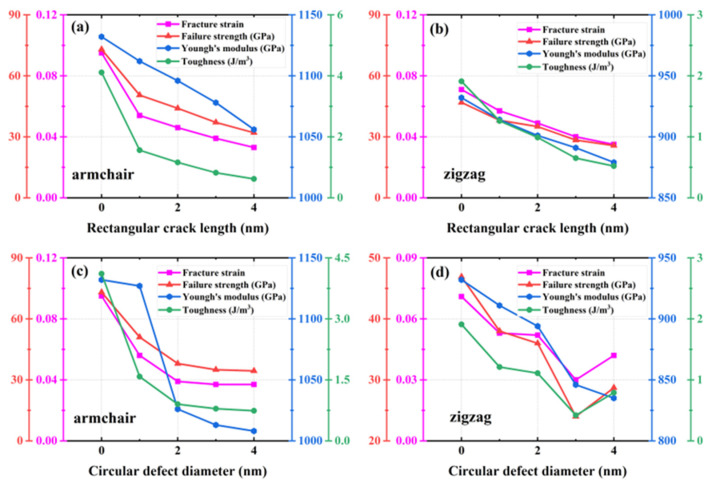
Four mechanical properties of HOP–graphene with rectangular cracks of different lengths under stretching (**a**) in the armchair direction and (**b**) in the zigzag direction. Four mechanical properties of HOP–graphene with circular cracks of different diameters under stretching (**c**) in the armchair direction and (**d**) in the zigzag direction.

**Figure 13 nanomaterials-15-00031-f013:**
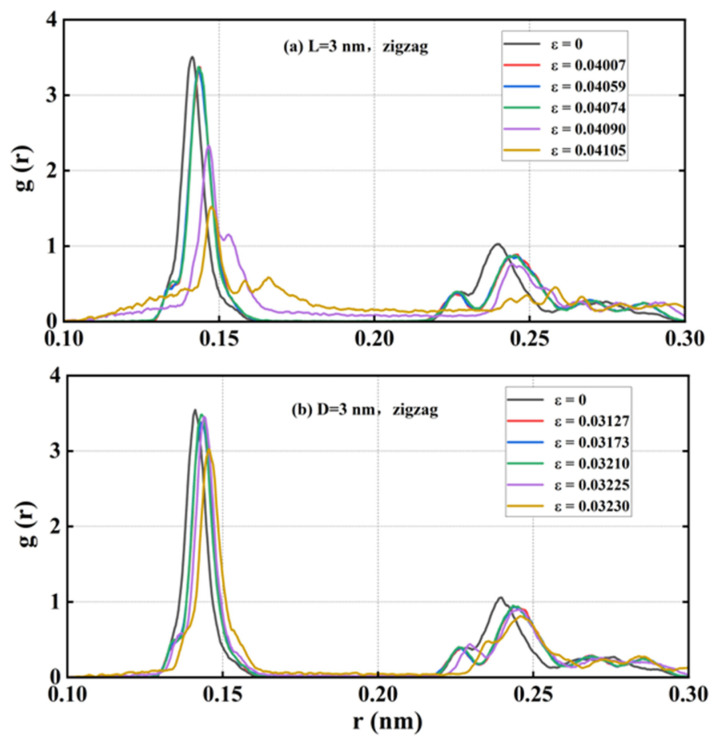
RDF with rectangular crack defects of 3 nm in length and circular hole defects of 3 nm in diameter when stretched in a zigzag direction.

**Table 1 nanomaterials-15-00031-t001:** Mechanical properties of HOP–graphene.

Tensile Direction	Fracture Strain	Faliure Strength (GPa)	Young’s Modulus (GPa)	Toughness (J m^−3^)
Along armchair	0.095	73.1	1132	4.11
Along zigzag	0.071	46.9	932	1.91

## Data Availability

The datasets generated during and/or analyzed during the current study are available from the corresponding author on reasonable request.

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
