# Peer review of "Atomistic Study on the Mechanical Properties of HOP–Graphene Under Variable Strain, Temperature, and Defect Conditions"

_nanomaterials, 2024, doi:10.3390/nano15010031_

Round 1
Reviewer 1 Report (Previous Reviewer 2)
Comments and Suggestions for Authors
all my comments have been addressed successfully. The manuscript is ready for publication
Reviewer 2 Report (Previous Reviewer 3)
Comments and Suggestions for Authors
The manuscript can be published in present form
This manuscript is a resubmission of an earlier submission. The following is a list of the peer review reports and author responses from that submission.
Round 1
Reviewer 1 Report
Comments and Suggestions for Authors
In this study, various mechanical properties of HOP-Graphene are analyzed using molecular dynamics simulations. The research process is conducted meticulously, providing quantitative reliability in the results obtained. The systematic analysis results make this paper valuable to researchers interested in this material. However, the discussion on the obtained results feels somewhat insufficient. An advantage of using molecular dynamics simulations is the ability to elucidate atomic-level factors influencing mechanical properties. Including the following two points, if feasible, could enhance this paper further:
1. In Figure 5, why is there a reversal in the relative magnitude of mechanical properties for zigzag and armchair configurations between HOP-Graphene and pure Graphene?
2. The study investigates mechanical properties at high temperatures, varying vacancy volume fractions, and with or without defects. However, why does the degree of degradation differ between the zigzag and armchair configurations?
Additionally, there are two terms that are unclear. While a specialized researcher might understand them, it would be beneficial to provide detailed explanations for a broader audience:
3. Please define "the coefficients of variation."
4. What is the definition of "amorphous state"? The paper seems to imply that it is defined when the von Mises stress reaches a critical point. Please explain the rationale for this definition, with references to previous studies.
Reviewer 2 Report
Comments and Suggestions for Authors
The authors use molecular dynamics to apply strain to a graphene nanosheet model, investigating the effects of strain direction, temperature, surface area, and the presence of defects or vacancies on mechanical properties such as toughness, failure strength, Young’s modulus, and fracture strain. These mechanical properties are influenced by the strain direction but exhibit minimal dependence on the sheet size. All properties decrease with rising temperature and are further reduced as more vacancies or defects are introduced on the nanosheet.
Why is the simulation timestep set so small (0.1 fs), when other researchers typically use a timestep of 1-2 fs? Is it because the system evolves so quickly that it would become unstable or crash if a larger timestep were used?
Why are the simulations so brief (40 ps)? Can the authors ensure that their systems have reached equilibrium within this timeframe? I suggest that the calculated values for all properties be presented along with their standard deviations during the simulations.
Is the graphene sheet, prepared with its hexagonal, pentagonal, and heptagonal phases, ideally flat in its initial state, or does it show slight surface curvatures and ripples?
How are mechanical properties such as toughness, failure strength, Young’s modulus, and fracture strain calculated from the NVT simulations?
The variables discussed (toughness, failure strength, Young’s modulus, and fracture strain) seem to be almost linearly correlated in all figures. Could you explain the reasons behind this correlation?
What is the computational cost in terms of nanoseconds per hour for the different strain rates tested in figure 4? What is the computational cost of simulating the nanosheet with 2,550 carbon atoms compared to those with 20,160 atoms and 40,800 atoms?
In the strain analysis of structures with defect configurations, only the actual stress and stress distribution contours are shown. Other values, such as Young’s modulus and failure strength, are discussed but not supported by figures. Please provide a comprehensive list of these properties for the various shapes and sizes of defects on the nanosheets.
Reviewer 3 Report
Comments and Suggestions for Authors
Referee report
Nanomaterials
Manuscript ID: nanomaterials-3304279
Atomistic study on the mechanical properties of HOP-Gra-phene under variable strain, temperature, and defect conditions
by Q. Peng, J. Li, X. Cai, G. Chen, Z. Huang, L. Zhen, H. Li, X.-J. Chen, Z. Hu
The paper is devoted to theoretical molecular-dynamics study of HOP (H(exagon)O(ctagon)P(entagon)) graphene consisting of 5-, 6- and 8-membered carbon rings theoretically predicted Mandal et. al. back in 2013 (this is important for further consideration of the manuscript). Using periodic boundary conditions in x- and y-directions (p. 2 of the Manuscript), molecular-dynamics study of the effects of varying sizes, strain rates, temperatures and defects on the mechanical properties were studied. It was found that Young's modulus of HOP-graphene in the armchair direction is 21.5% higher than that in the zigzag one. The increase in temperature from 100 K to 900 K resulted in a decrease in Young's modulus by 7.8% and 2.9% for stretching along the armchair and zigzag directions, respectively. Introduction of void defects from 0 to 3% resulted in in decrease of Young’s moduli by 24.7 and 23.1%, respectively. The increase in the length of rectangular crack defects from 0 nm to 4 nm resulted in a decrease in the Young's modulus for stretching along the armchair and zigzag directions by 6.7% and 5.7%, respectively. The influence of a circular hole (up to 4 nm) also resulted in decrease of the moduli up to 11.0%.
In fact, introduction of periodic boundary conditions in x- and y- directions with just 10 carbon atoms in the unit cell (Figure 1a, p. 3 of the manuscript) may impose critical limitations on topology and symmetry of 2D HOP-graphene regular atomic lattice. Just after the introduction of HOP-graphene by Mandal et. al. in 2013, the structure and properties of numerous 2D carbon-based perfect crystalline lattices constituted by 4-, 5-, 6-, 7-, and 8-member rings were investigated by several authors (see below) using theoretical simulations at different levels of theory (see numerous publications 2015 – 2022). Right now it is well known that such lattices prone to mechanical (https://doi.org/10.1016/j.flatc.2016.12.001), thermodynamic (https://doi.org/10.1016/j.carbon.2015.09.092), symmetrical and topological (https://doi.org/10.1021/acs.jpclett.5b02309;https://doi.org/10.1039/D0CP00979B), energetic and kinetic (https://doi.org/10.1073/pnas.1520402112 ) instability.
It was shown that perfectly planar complex 2D lattices may accumulate mechanical stress generated by structural mismatch of different inequivalent sublattices. Periodic Boundary Conditions may artificially stabilize planar 2D crystalline lattices by compensating the stress through TI symmetrical restrictions. Since at least one force constant perpendicular to the plane is effectively equal to 0, it may cause strong topological instability of low-dimensional crystal lattices with formation of wave-like bend structures or even irregular atomic clusters. In fact, introduction of Periodic Boundary conditions for complex low-dimensional crystalline lattices is equivalent to combination of infinitely large positive pressure and stretching the lattice in both x- and y- directions.
It is necessary to note that some unstable materials can be stabilized by a suitable substrate or by creation of finite-sized flakes of limited dimensions. It was found that the absence of imaginary modes in phonon spectra cannot be used as solid and final proof of structural stability of such lattices.
Another critically important aspect is that it was found that Crystallography Restriction Theorem cannot be used to analyze low-dimension crystalline lattices with multiple non-equivalent sublattices. Instead, Topology Conservation Theorem should be used to analyze structure and symmetry of low-dimensional crystals with multiple non-equivalent sublattices. It is obvious that HOP-graphene does not satisfy mandatory TCP requirements and should form some bended structures like structural waves.
I’m pretty sure that 2D planar structure of HOP-graphene is just a result of erroneous application of periodic boundary conditions of very limited dimensions with 5.647´4.908Å 10 carbon atom primitive unit cell. Strongly anisotropic mechanical properties (Young’s moduli in x- and y- directions, see above) of the lattice directly prove this conclusion. It seems that introduction of proportionally larger unit cells (from 5´5 or 10´10 unit cells) may result in formation of structural waves of different periodicity (up to hundreds Å) with completely different mechanical properties.
To complete the study the authors of the manuscript should prove:
1. HOP-graphene lattice can exist and it is topologically and quantum stable;
2. Reveal the true structure of the lattice. Following TCT and quantum stability mandatory requirements, probably it can exist only in the form of small wave-like flakes;
3. Reconsider the mechanical properties and response of the lattice on temperature and different defects.
At this stage I strongly recommend the Editorial Board of Nanomaterials to reject the manuscript as completely erroneous.
--------------
Disclaimer: All cited publications are included in the report for reference only and do not imply endorsement or authorship by the reviewer. The reviewer does not require citation of these publications in the revised version of the manuscript.